Exploring molecular variation and combining ability of local and exotic bread wheat genotypes under well-watered and drought conditions

http://orcid.org/0000-0002-5278-0104 Motawei Mohamed I. 1 mtaoa@qu.edu.sa
Kamara Mohamed M. 2
Rehan Medhat 1 m.rehan@qu.edu.sa
1 Department of Plant Production, College of Agriculture and Food, Qassim University , Buraydah , Saudi Arabia
2 Department of Agronomy, Faculty of Agriculture, Kafrelsheikh University , Kafr El-Sheikh , Egypt
Kutlu Imren
Electronic publication date: 2025 Feb 27
Publication date: 2025
Volume: 13
Electronic Location ID: e18994
Received 2024 Nov 20; Accepted 2025 Jan 23
Copyright: © 2025 Motawei et al.
Copyright year: 2025
Copyright holder: Motawei et al.
License: This is an open access article distributed under the terms of the Creative Commons Attribution License, which permits unrestricted use, distribution, reproduction and adaptation in any medium and for any purpose provided that it is properly attributed. For attribution, the original author(s), title, publication source (PeerJ) and either DOI or URL of the article must be cited.
License URL: https://creativecommons.org/licenses/by/4.0/

Keywords: Drought, SSR markers, Molecular diversity, Combining ability, Climate-adaptive breeding, Sustainability, Diallel mating design, Combining ability, Bread wheat

Funding: Qassim University 2023-SDG-l-BSRC35790 This work was supported by Qassim University, represented by the Deanship of Scientific Research, number 2023-SDG-l-BSRC35790 during the academic year 1445 AH/2023 AD. The funders had no role in study design, data collection and analysis, decision to publish, or preparation of the manuscript.

==============================
Drought is one of the most environmental stressors, significantly affecting wheat production, particularly in the face of accelerating climate change. Therefore, developing drought-resistant, high-yielding wheat varieties is essential to ensure sustainable production and maintain global food security as the world population rapidly grows. This study aimed to evaluate the genetic variation of local and imported bread wheat genotypes through simple sequence repeat (SSR) markers and assess their combining ability to identify top-performing genotypes under both normal and drought-stress environments. SSR markers revealed significant genetic diversity among the parental genotypes, which were utilized to develop 28 F1 crosses utilizing diallel mating design. Field trials under well-watered and drought-stressed environments demonstrated that drought significantly reduced all measured agronomic traits. The genotypes were categorized into five clusters based on their drought tolerance, ranging from highly sensitive (group-E) to robustly drought-resistant (group-A). The local variety Sids-12 (P2) was identified as an excellent combiner for breeding shorter and early-maturing cultivars and Line-117 (P3), Line-144 (P4), and Line-123 (P5) for improving grain yield and related traits under drought conditions. The crosses P1×P5, P3×P8, P4×P5, and P6×P7 possessed superior performance under both conditions. Key traits, including plant height, grains per spike, 1,000-grain weight, and spikes per plant, displayed strong correlations with grain yield, providing an effective approach for indirect selection in drought-prone environments.

Introduction

Wheat (Triticum aestivum L.) is one of the most significant cereal crops globally (Li et al., 2024b). It provides a vital source of calories, carbohydrates, protein, and vitamins for human consumption (Khalid, Hameed & Tahir, 2023). Additionally, its straw is extensively used for animal feed and in various industrial applications (Taghizadeh-Alisaraei et al., 2023). Climate change is a major global challenge to agricultural production and food security. Projected increases in temperature and unpredictable rainfall patterns are expected to worsen and occur more frequently (Ishaque et al., 2023). Drought stress is a major abiotic factor that presents a serious risk to wheat production worldwide (Mannan et al., 2022; Thabet et al., 2024). The decline in yearly crop yields across the world due to drought is greater than the total reduction caused by all other factors. Water deficiency significantly hampers plant growth, development, and yield (Ali, Mansour & Awaad, 2021; Sewore, Abe & Nigussie, 2023). Nearly half of the global cultivated wheat area is subjected to drought stress. Additionally, increasing urbanization and industrialization have intensified the demand for water supplies (Cramer et al., 2018). Consequently, water scarcity and drought issues are projected to escalate, posing further challenges to wheat production. North Africa ranks among the driest and most drought-stricken regions globally due to climate change (Hamed et al., 2018). In Egypt, the limited average annual water availability poses a significant challenge for wheat production, contributing to a severe water scarcity issue that threatens agricultural sustainability (Abdelaal & Thilmany, 2019). As a country within the Mediterranean basin, Egypt could experience a reduction in wheat grain yield by 18% to 24% because of climate change (Morsy et al., 2022).

Breeding genotypes that are tolerant to drought is among the most effective strategies for alleviating the adverse effects of drought on wheat yield in drought-prone environments (Kamara et al., 2022; Morsi et al., 2023). Therefore, enhancing tolerance and adaptability to drought stress has become a primary goal of breeding programs (Al-Ashkar, 2024; Gaballah et al., 2024; Sedhom et al., 2024). Current efforts are concentrated on developing high-yielding, drought-resistant wheat genotypes, which is particularly critical given the challenges of climate change (Ezzat et al., 2024). However, progress in enhancing drought tolerance is impeded by insufficient knowledge regarding the potential of existing genetic resources. This limitation can be overcome by evaluating the combining abilities of available genetic resources under drought conditions. Combining ability includes general (General Combining Ability; GCA) for parents and specific (Specific Combining Ability; SCA) for hybrids (ElShamey et al., 2022; Heiba et al., 2023). The diallel mating design is a valuable approach for studying combining abilities effects and identifying the type of gene action governing trait inheritance. GCA primarily reflects additive gene effects, while SCA is related to non-additive effects (Kamara et al., 2024). By assessing GCA and SCA, researchers can identify the most effective parental genotypes and crosses that exhibit significant hybrid vigor in drought-stressed environments (Galal et al., 2023). Previous research has successfully utilized diallel mating to pinpoint high-performing parents and hybrids, leading to advancements in stress tolerance traits in bread wheat (Chaudhari et al., 2023; Farshadfar, Rafiee & Hasheminasab, 2013; Kaur & Kumar, 2024; Semahegn et al., 2021).

Assessing genetic diversity is crucial for the effective characterization and preservation of germplasm (Ahmed et al., 2023; Zannat et al., 2023). Diversity encompasses differences in morphological and agronomic traits, as well as molecular markers that indicate genomic variations (Gracia et al., 2012; Ponce-Molina et al., 2012). Simple sequence repeats (SSRs), a type of molecular marker, are used to detect DNA variation across genotypes (Bidyananda et al., 2024). SSRs, short nucleotide tandem repeats (2–5 bp), are abundant and codominant markers, and widely distributed across the genome (Zhao et al., 2023). SSRs have been instrumental in cultivar characterization, differentiating genetic resources, and implementing marker-assisted selection across a range of plant species (Atsbeha et al., 2024; Su et al., 2023). SSR markers, known for their high polymorphism, reproducibility, and co-dominant inheritance, have proven to be a powerful tool for assessing genetic diversity and identifying drought-resilient genotypes (Belete et al., 2021; Mammadova et al., 2024a). Research has demonstrated that SSR markers can effectively differentiate wheat genotypes based on their tolerance to drought stress by targeting specific genomic regions associated with drought-responsive traits (Haque et al., 2021; Mammadova et al., 2024b). Additionally, SSR markers are employed to facilitate the identification of quantitative trait loci (QTL) linked to drought tolerance (Ahmed et al., 2023; Ezzat et al., 2024; Shukla et al., 2021). Therefore, enabling precise selection of drought-tolerant parents for hybridization, SSR-based analyses could contribute to breeding programs aimed at developing high-yielding, water-efficient wheat cultivars (Li et al., 2024a; Singh et al., 2016; Verma, Borah & Sarma, 2019). This study aimed to assess the genetic variation of local and imported bread wheat genotypes through SSR markers, estimate GCA, SCA, and heterotic effects, and analyze trait relationships in both well-watered and drought stress environments.

Materials and Methods

Plant material

This study utilized eight bread wheat (Triticum aestivum L.) genotypes (Table S1). These included three widely cultivated local Egyptian varieties, known for their adaptation to the region environmental conditions, and five advanced lines imported from CIMMYT (International Maize and Wheat Improvement Center). The local cultivars were chosen for their agronomic relevance and performance under traditional farming practices in Egypt; the CIMMYT lines were selected based on their pedigree and reported contributions to global wheat improvement efforts, particularly in enhancing drought tolerance. Together, these genotypes provide a robust framework for evaluating genetic and phenotypic variability under contrasting irrigation regimes.

Molecular profiling

DNA was extracted from about 50 mg of fresh young leaf samples collected from seedlings at the 3-week growth stage from regarded genotypes following the cetyltrimethylammonium bromide (CTAB) method (Doyle, 1990). The purity and quantity of the extracted DNA were evaluated using NanoDrop spectrophotometer (ND-1000 Spectrophotometer). Twelve markers were utilized in this study, selected from prior research due to their reliable correlation with drought tolerance in wheat. The sequences of the primers used are listed in Table 1. The SSR markers were chosen from previously published reports due to their association with wheat drought tolerance (Ateş-Sönmezoğlu, Çevik & Terzi-Aksoy, 2022; Ateş Sönmezoğlu & Terzi, 2018; Belete et al., 2021). The PCR technique reaction contained 10 µl master mix (DreamTaq PCR Master Mixes (2X), 1 μL of genomic DNA (20 ng/μL), and 0.5 μM of each primer. The PCR reaction began with 2 min for an initial denaturation at 94 °C, followed by 35 cycles of denaturation at 94 °C for 50 s, annealing for 30 s at 55–60 °C, and extension for 30 s at 72 °C. It concluded with a final elongation at 72 °C for 7 min. Subsequently, amplification results were analyzed through electrophoresis on a 1.5% agarose gel. A binary data matrix, scored with values of 0 and 1 based on the presence or absence of amplified bands for each SSR marker, was generated. The number of total bands (TB), monomorphic (MB) and polymorphic bands with percentages were calculated. Genetic distances were calculated using PAST software, and a dendrogram was constructed using the unweighted pair group method with arithmetic mean (UPGMA) in the Multi-Variate Statistical Package (MVSP) software package version 3.1.

Table 1 The simple sequence repeat (SSR) markers utilized.

Markers	Forward primer	Reverse primer	
Xg wm-626	GATCTAAAATGTTATTTTCTCTC	TGACTATCAGCTAAACGTGT	
Xg wm-603	ACAAACGGTGACAATGCAAGGA	CGCCTCTCTCGTAAGCCTCAAC	
Xg wm-484	ACATCGCTCTTCACAAACCC	AGTTCCGGTCATGGCTAGG	
Xg wm-389	ATCATGTCGATCTCCTTGACG	TGCCATGCACATTAGCAGAT	
Xg wm-357	TATGGTCAAAGTTGGACCTCG	AGGCTGCAGCTCTTCTTCAG	
Xg wm-337	CCTCTTCCTCCCTCACTTAGC	TGCTAACTGGCCTTTGCC	
Xg wm-186	GCAGAGCCTGGTTCAAAAAG	CGCCTCTAGCGAGAGCTATG	
Xg wm-99	AAGATGGACGTATGCATCACA	GCCATATTTGATGACGCATA	
Xg wm-11	GGATAGTCAGACAATTCTTGTG	GTGAATTGTGTCTTGTATGCTTCC	
Xw mc-89	ATGTCCACGTGCTAGGGAGGTA	TTGCCTCCCAAGACGAAATAAC	
Xw mc-78	AGTAAATCCTCCCTTCGGCTTC	AGCTTCTTTGCTAGTCCGTTGC	
Xp sp-3200	GTTCTGAAGACATTACGGATG	GAGAATAGCTGGTTTTGTGG	

Generation of F1 crosses and field evaluation

An 8 × 8 half-diallel mating scheme, was employed to generate 28 F1 crosses in the 2021–2022 season. Hybrid grains were developed through hand emasculation and pollination techniques. During 2022–2023 season, the parents and their F1 offspring were assessed under two irrigation regimes at the Kafrelsheikh University Experimental Farm (31°6′N, 30°56′E), Egypt. The experimental site is characterized by an arid climate, with minimal annual rainfall averaging 55 mm, as depicted in Fig. 1. During the growing season, average daytime temperatures ranged from 18 °C to 36 °C, with occasional peaks exceeding 40 °C, while nighttime temperatures varied from 8 °C to 22 °C. Relative humidity levels were consistently low, averaging 40–75%. The physical and chemical properties of the soil at the experimental site are provided in Table S2. The soil analysis revealed that the soil is clay with a profile comprised of 53.7% clay, 32.3% silt, and 14% sand. The electrical conductivity and pH were 3.59 dS/m and 8.42. The field capacitance of 35.1% and a persistent wilting threshold of 19.8% were among the soil water characteristics. A Randomized Complete Block Design in three replicates was used for each treatment. A 6-meter-wide space was left between the two irrigation treatments to stop water from leaking through. The well-watered condition involved five irrigation events throughout the entire season, totaling 4,380 m3/ha, while the drought stress condition included two irrigation events, amounting to 2,860 m3/ha. In both treatments, the first irrigation was applied during the establishment phase to ensure uniform seedling growth. For the drought stress condition, the second and final irrigation was applied at the tillering stage, after which no additional water was supplied. This created a water deficit stress environment during the subsequent critical stages, including heading, flowering, and grain filling, which are known to have the greatest impact on yield and quality under water-limited conditions. In the well-watered treatment, additional irrigations were applied at the heading, flowering, and grain-filling stages to ensure optimal soil moisture throughout the season. This contrast in irrigation timing and quantity between the two treatments allowed for a clear assessment of the genotypic responses to drought stress during these critical growth phases. Plots included three 3 m long rows, spaced 0.30 m apart between rows and 0.15 m between plants. Fertilizers were applied at rates of 180 kg/ha for nitrogen (N), 57 kg/ha for potassium (K2O), and 35 kg/ha for phosphorus (P2O5).

Figure 1 Meteorological characterization for the growing winter seasons.

Data collection

Days to heading were noted as the count from planting until 50% of spikes emerged. At physiological maturity, plant height was quantified by measuring the length from the soil surface to the tip of the spike in centimeters. Spike length (cm), and number of grains/spikes were assessed on 10 randomly selected spikes from each plot. A total of 1,000-grain weight was determined by weighing 1,000 grains in grams. Grain yield was calculated by harvest ten guarded plants within each plot, then drying, threshing, and recording the total grain yield per plant in grams.

Statistical analysis

Data was analyzed using ANOVA in R (version 4.2.2), applying the least significant difference (LSD) test at P < 0.05 and P < 0.01 to assess mean differences. Combining abilities were computed using Griffing’s Method 2, Model 1 (Griffing, 1956) employing the following model: xij = µ + ĝi + ĝj + ŝij + eijkl. Where; xij represent the observed value of the cross between parent i and parent j, µ is the population mean, ĝi and ĝj are GCA effect ith and jth parents, ŝij is SCA effect for the cross between i and j parent, eijkl denotes the environmental effect specific to the jkl observation. The GCA/SCA ratio was calculated by comparing the mean squares of GCA and SCA. Differences between GCA and SCA effects were determined using the standard error and the tabulated t-value. Cluster analysis, principal component, and heatmap analyses were conducted using R programming version. FactoMiner package was used for principal component analysis (PCA), factoextra was used for cluster analysis, and pheatmap was used for heatmap with clustering. Better-parent (BP) heterosis was calculated as follows

Betterparent(BP)heterosis=F1−BPBP×100.

Using grain yield under normal (Yp) and drought stress (Ys) conditions, the Stress Susceptibility Index (SSI) was calculated following Fischer & Maurer (1978) through the equation below:

SSI=[(1−(Ys/Yp)][1−(Ys/Yp)]×100.

Results

Molecular analysis

Twelve SSR markers were used to evaluate genetic diversity in the assessed parental genotypes, detecting 22 polymorphic allele/bands with 1 to 5 bands per primer with an average of 1.83 bands (Table 2). All the markers exhibited 100% polymorphism, except for Xgwm 626, resulting in an overall average polymorphism of 98.33%. The genetic distances measured using SSR markers varied between 0.13 and 0.83, with an average of 0.53 (Table 3). The smallest genetic distance was recorded between P5 and P8, while the largest genetic distance was noted between P6 and P7.

Table 2 The total number of bands (TB), along with monomorphic (MB) and polymorphic bands produced by the SSR markers applied in this study.

Primers	TB	MB	PB	Polymorphism (%)	
Xgwm-11	1	0	1	100	
Xgwm-99	1	0	1	100	
Xgwm-186	2	0	2	100	
Xgwm. 337	1	0	1	100	
Xgwm. 357	3	0	3	100	
Xgwm. 389	2	0	2	100	
Xgwm. 484	1	0	1	100	
Xgwm. 603	1	0	1	100	
Xgwm. 626	5	1	4	80	
Xgwm-3200	3	0	3	100	
Xgwm-78	2	0	2	100	
Xgwm-89	1	0	1	100	
Total	23	1	22		
Average	1.92	0.08	1.83	98.33	

Table 3 Genetic distance between the assessed wheat genotypes based on SSR markers.

Parent	P 1	P 2	P 3	P 4	P 5	P 6	P 7	P 8	
P. 1 (Gemmeiza-12)	–								
P. 2 (Sids-12)	0.59	–							
P. 3 (Line-117)	0.47	0.40	–						
P. 4 (Line-144)	0.61	0.18	0.35	–					
P. 5 (Line-123)	0.36	0.39	0.28	0.42	–				
P. 6 (Line-125)	0.77	0.43	0.48	0.38	0.55	–			
P. 7 (Gemmeiza-7)	0.80	0.80	0.82	0.81	0.79	0.83	–		
P. 8 (Line-121)	0.47	0.47	0.37	0.50	0.13	0.48	0.79	–	

Based on their similarity coefficients, the dendrogram (Fig. 2) shows a hierarchical clustering of eight wheat parents (P1 to P8). The hierarchical clustering analysis revealed two distinct groups. The first group involved two clusters: Cluster 1 contained parents P1, P5, and P8 at a relatively high similarity coefficient (~0.10). This suggests a strong relationship among these parents, indicating that they share significant similarities. Cluster 2 enclosed P3 that joined Cluster 1 (P1, P5, P8) at a lower similarity coefficient (~0.13), suggesting a moderate relationship with these genotypes with some common characteristics. Furthermore, group 2 included two clusters: Cluster 3 comprised of parents P2 and P4 and formed a separate sub-cluster, with a similarity coefficient of approximately 0.07, implying that they share a high degree of similarity. Cluster 4 included P6 at a low similarity coefficient with cluster 3 reached (~0.05), whereas P7 considered as an out of group parent.

Figure 2 Dendrogram for clustering eight wheat genotypes based on SSR markers.

Analysis of variance

The results in Table 4 revealed a highly significant difference (P ≤ 0.001) between the evaluated genotypes, parents, F1 crosses, and parents vs. crosses across in most traits under both irrigation regimes. Partitioning the genotypes into GCA and SCA components demonstrated that the GCA and SCA were highly significant (P ≤ 0.001) for all assessed traits under both normal and drought environments. The GCA/SCA ratio was below unity for all traits across both irrigation regimes, except for plant height under drought stress conditions.

Table 4 Mean squares for all investigated traits from ordinary and combining ability analysis under normal (NOR) and drought (WAD) conditions.

Source of variance	DF	Days to heading	Plant height (cm)	Spike length (cm)	Number of spikelets/spike	
NOR	WAD	NOR	WAD	NOR	WAD	NOR	WAD	
Genotype	35	11.52 **	12.54 **	136.23 **	197.74 **	2.12 **	2.95 **	4.13 **	6.63 **	
Parent	7	28.17 **	22.76 **	106.53 **	184.06 **	3.35 **	4.82 **	4.79 **	5.24 **	
F1 Cross	27	7.63 **	10.35 **	147.52 **	207.86 **	1.77 **	2.55 **	3.76 **	6.48 **	
Parent vs. Cross	1	0.07	0.15	39.28	20.40	2.90 *	0.60	9.69 **	20.56 **	
Error	70	3.00	4.15	20.30	19.88	0.48	0.82	1.07	0.70	
GCA.	7	6.03 **	7.94 **	125.58 **	223.03 **	1.83 **	2.69 **	1.56 **	1.66 **	
SCA.	28	3.29 **	3.24 **	25.37 **	26.63 **	0.42 **	0.56 **	1.33 **	2.35 **	
Error	70	1.00	1.38	6.77	6.63	0.16	0.27	0.36	0.23	
GCA./SCA.		0.22	0.35	0.64	1.08	0.64	0.86	0.12	0.07	
Source of variance	DF	Number of spikes/plant	Number of grains/spikes	1,000-grain weight (g)	Grain yield/plant (g)	
NOR	WAD	NOR	WAD	NOR	WAD	NOR	WAD	
Genotype	35	14.53 **	17.35 **	36.84 **	60.46 **	32.49 **	45.26 **	63.76 **	104.73 **	
Parent	7	8.84	11.99 *	28.72 **	24.17 **	25.64 *	46.33 **	43.83 **	81.81 **	
F1 Cross	27	16.45 **	18.79 **	38.99 **	68.76 **	31.71 **	46.48 **	69.13 **	110.30 **	
Parent vs Cross	1	2.74	16.07	35.62 *	90.54 **	101.40 **	4.68	58.21 *	114.62 **	
Error	70	4.97	4.48	5.71	7.28	10.62	13.74	13.15	14.45	
GCA.	7	11.45 **	15.87 **	7.72 **	24.67 **	12.56 **	19.07 **	35.68 **	80.84 **	
SCA.	28	3.19 *	3.26 **	13.42 **	19.03 **	10.40 **	14.09 **	17.64 **	23.43 **	
Error	70	1.66	1.49	1.90	2.43	3.54	4.58	4.38	4.82	
GCA./SCA.		0.64	0.81	0.05	0.13	0.13	0.15	0.24	0.41	
Note:

* and ** imply significance at 0.05 and 0.01, respectively, with the corresponding values bolded for clarity.

Performance of the evaluated parents and f1 crosses

The generated F1 crosses, and their parents demonstrated considerable variability in all traits under both conditions. Under drought stress, all genotypes accelerated their heading by an average of 3.16 days in comparison to the well-watered conditions. P2 and P4×P7 had the earliest heading, where P4 and P3×P5 displayed the latest heading across the two treatments (Fig. 3A). Plant height was significantly reduced by 10.0% under stressed conditions compared to normal conditions. The shortest plant heights were observed in P2 and P3 parents and P1×P7 and P2×P4 crosses. In contrast, the tallest plants were found in parents P7 and P8, along with the crosses P6×P7 and P5×P6 under both irrigation treatments (Fig. 3B). Water deficit treatment significantly decreased spike length by 4.04% under drought stress. The longest spike was assigned for the parents P1 and P2, and the crosses P4×P7 and P4×P8 under normal conditions, while the parents P1 and P4 and the crosses P2×P4, and P4×P7 recorded the highest ones under stress conditions (Fig. 3C).Similarly, the number of spikelets per spike reduced by 6.78% under stress conditions. The parent P1, and the crosses P4×P7, P4×P8, P2×P7, and P5×P6 recorded the highest number of spikelets (Fig. 3D). Spikes per plant also declined by 20.39% due to drought stress. The highest means of spikes per plant were found in parents P3 and P5, and crosses P4×P6 and P3×P8 under normal conditions. In contrast, under stress conditions, parents P5 and P8, and crosses P3×P6 and P3×P8, exhibited the highest values (Fig. 4A). Similarly, the number of grains per spike decreased significantly by 10.91% under stress treatment. P1, P4, P4×P5, and P3×P8, showed the greatest grains per spike under both stressful and normal conditions (Fig. 4B). The 1,000-grain weight was significantly affected by deficit irrigation, decreasing by 9.79% compared to normal irrigation. P7 and P8 as well as P1×P8, and P6×P8 crosses give the maximum weight under well-watered conditions, while P1 and P8, and P3×P8, and P7×P8 crosses possessed the heaviest weight under drought stress treatment (Fig. 4C). Drought stress severely impacted grain yield, causing a 23.41% reduction compared to normal conditions. The highest grain yields under normal conditions were achieved by parental genotypes P3 and P4 and the crosses P1×P5 and P3×P8. Under stress conditions, the highest grain yield was assigned for the parents P4 and P8, and the crosses P4×P5 and P3×P8 (Fig. 4D).

Figure 3 Performance of the parental and their 28 wheat F1s genotypes.

(A) Days to heading; (B) plant height; (C) spike length; and (D) number of spikelets per spike.

Figure 4 Performance of the parental and their 28 F1s genotypes.

(A) Number of spikes per plant; (B) number of grains per spike; (C) 1,000 grain weight; and (D) grain yield per plant.

Classification of evaluated genotypes

The genotypes were classified based on drought tolerance using hierarchical clustering analysis. The drought tolerance indices were calculated for each genotype based on grain yield under drought-stressed and well-watered conditions. These indices were employed to group assessed genotypes based on their tolerance to drought tolerance. The clustering was performed using Ward’s method with Euclidean distance as the metric to ensure a clear distinction between clusters. The thirty-six genotypes were grouped into five clusters according (Fig. 5). Group A consisted of four genotypes (P1×P5, P5×P6, P6×P8 and P7×P8) that exhibited the highest drought tolerance indices. Consequently, they are regarded as highly drought-tolerant genotypes. Likewise, Group B also included nine genotypes (P6, P8, P3×P8, P5×P7, P4×P7, P4×P6, P3×P7, P4×P8 and P1×P3) with intermediate-high values. As a result, they are designated as moderately tolerant. Similarly, Group C contained 13 genotypes with intermediate-low drought tolerance indices. Thus, they are deemed as moderately sensitive. Conversely, Group D (six genotypes) and E (four genotypes), had the lowest stress sensitivity. Thus, they are classified as drought-sensitive and highly sensitive genotypes, respectively. This grouping aids in selecting genotypes for improving drought tolerance.

Figure 5 Dendrogram based on stress susceptibility index of 36 wheat genotypes (eight parental genotypes and their 28 F1s).

GCA effects

Positive GCA effects are favorable for all assessed characters, except heading date, and plant height, where negative values are more desirable. As indicated in Table 5, P1 showed significantly negative GCA effects for plant height under both conditions, while its effects for other traits were insignificant or undesirable. P2 was identified as a superior general combiner for both dwarf stature and early maturity under normal and stress conditions. Additionally, P3 presented significant GCA effects for the number of spikes/plant and grain yield under both normal and stressful conditions. Likewise, P4 demonstrated highly significant positive effects for the number of grains per spike under stressed conditions, along with number of spikelets per spike and spike length under both conditions. Line P5 displayed significantly the uppermost GCA for grain yield under both treatments. Line P6 was identified as a poor combiner for most of the traits studied, exhibiting undesirable effects, whether significant or insignificant. P7 showed significant negative effects for days to heading in well-watered conditions, along with positive effects for spike length in both conditions. Line P8 was an excellent combiner for number of spikes per plant, 1,000-grain weight, number of grains per spike in both conditions, and grain yield under drought stressconditions.

Table 5 Estimated general combining ability (GCA) effects of eight bread wheat parental genotypes for assessed traits under normal (NOR) and drought (WAD) conditions.

Parent	Days to heading	Plant height (cm)	Spike length (cm)	Number of spikelets per pike	
NOR	WAD	NOR	WAD	NOR	WAD	NOR	WAD	
P1 (Gemmeiza-12)	0.30	0.65	−5.14**	−5.99**	0.15	0.22	0.26	0.18	
P2 (Sids-12)	−1.47**	−1.85**	−5.21**	−8.20**	0.18	0.15	0.27	−0.35*	
P3 (Line-117)	0.73*	0.85*	−1.51	−1.50	−0.64**	−0.80**	−0.67**	−0.61**	
P4 (Line-144)	0.80**	0.08	1.22	2.63**	0.46**	0.65**	0.37*	0.70**	
P5 (Line-123)	0.43	0.22	1.46	2.47**	−0.13	−0.15	−0.51**	−0.09	
P6 (Line-125)	−0.40	−0.68	3.49**	3.80**	−0.63**	−0.61**	−0.02	−0.23	
P7 (Gemmeiza-7)	−0.63*	0.05	2.93**	3.69**	0.39**	0.60**	0.34	0.20	
P8 (Line-121)	0.23	0.68	2.77**	3.11**	0.21	−0.06	−0.04	0.20	
LSD.(0.05)gi	0.59	0.69	1.53	1.52	0.24	0.31	0.35	0.28	
LSD.(0.01)gi	0.78	0.92	2.03	2.01	0.31	0.41	0.47	0.38	
Parent	No. of spikes per plant	No. of grains per spike	1,000-grain weight (g)	Grain yield per plant (g)	
NOR	WAD	NOR	WAD	NOR	WAD	NOR	WAD	
P1 (Gemmeiza-12)	−0.73	−0.07	0.77	0.48	−0.36	0.78	−1.19	−0.86	
P2 (Sids-12)	−1.81**	−2.49**	−0.69	−2.38**	−0.05	−2.37**	−1.96**	−5.21**	
P3 (Line-117)	1.46**	1.56**	−1.16**	−0.95*	−1.66**	−1.24	1.35*	1.43*	
P4 (Line-144)	0.10	−0.06	0.69	3.02**	0.51	0.24	3.00**	3.25**	
P5 (Line-123)	1.01**	0.70	−0.59	−0.45	−0.94	0.41	1.45*	1.29*	
P6 (Line-125)	−0.31	0.49	−0.69	−0.08	0.40	−0.68	−1.46*	1.05	
P7 (Gemmeiza-7)	−0.54	−0.99**	0.47	−0.55	−0.04	0.80	−2.00**	−3.01**	
P8 (Line-121)	0.81*	0.86*	1.21**	0.92	2.14**	2.06**	0.81	2.06**	
LSD (0.05) gi	0.76	0.72	0.81	0.92	1.11	1.26	1.23	1.29	
LSD (0.01) gi	1.00	0.95	1.08	1.22	1.47	1.67	1.64	1.71	
Note:

* and ** imply significance at 0.05 and 0.01, respectively, with the corresponding values bolded for clarity.

SCA effects

The evaluated crosses showed diverse SCA effects across all the traits studied under normal and stressed environments (Table 6). The crosses P4×P7 and P6×P8 exhibited significant negative SCA effects for days to heading under both treatments, making them useful for improving earliness in wheat breeding. Similarly, favorable negative effects for plant height were observed in P1×P7, P1×P6, P2×P4, P5×P8, and P1×P8 under drought and well-watered conditions. Otherwise, the uppermost significantly positive SCA effects for spike length were detected in crosses P3×P5 and P4×P6 under well-watered conditions, and P3×P5 and P5×P6 under drought conditions. Additionally, the hybrid P5×P6 had the largest positive effects for the spikelets number per spike under well-watered conditions, whereas P1×P3, P3×P8, P4×P7, P4×P8, P5×P6, and P6×P7 demonstrated the strongest effects under drought stress. Crosses such as P3×P8, P4×P6, and P7×P8 under normal, and P1×P2, P3×P6, and P3×P8 under drought stressconditions presented the maximum positive values for the spikes number per plant. Likewise, the desirable SCA effect for number of grains per spike was noticed in P3×P8, P4×P5, and P6×P7 under both irrigation treatments. The crosses P1×P8, P1×P2, P2×P6, P3×P6, P3×P8, and P6×P8 displayed the highest significantly positive SCA values under normal irrigation for 1,000-grain weight, while P1×P2, P3×P8, P4×P5, P4×P7, and P7×P8 showed the highest effects under stress conditions. Regarding grain yield, the crosses; P2×P4 in normal conditions, P1×P2 and P3×P5 in stressed conditions, along with P3×P8, P1×P5, P4×P5, and P6×P7 under both conditions, displayed the uppermost significantly positive SCA effects. Remarkably, no hybrid showed advantageous SCA effects across all traits examined concurrently. Nevertheless, P3×P8 and P4×P5 showed positive effects on grain yield and contributing traits, proving effective as specific combinations under drought and well-watered conditions

Table 6 Specific combining ability effects of 28 F1 crosses for assessed traits under normal and drought conditions.

Cross	Days to heading	Plant height (cm)	Spike length (cm)	No. of spikelets/spike	
NOR	WAD	NOR	WAD	NOR	WAD	NOR	WAD	
P1×P2	−0.80	0.59	6.97**	9.40**	−0.43	−0.42	0.63	0.36	
P1×P3	−2.66**	−1.44	−2.05	−0.26	−0.64	−0.31	−0.17	1.05*	
P1×P4	0.27	0.66	−2.13	1.89	−0.51	−0.49	0.16	−0.31	
P1×P5	−1.03	−0.81	−0.85	−8.00**	−0.75*	−0.79	−0.13	−1.03*	
P1×P6	0.80	−1.24	−5.97*	−5.73*	−0.09	0.24	−0.95	−0.82	
P1×P7	1.70	2.36*	−8.99**	−7.28**	0.18	−0.37	−3.64**	−3.59**	
P1×P8	0.17	0.39	−5.14*	−8.38**	0.42	0.72	0.42	0.68	
P2×P3	0.10	1.06	3.53	3.46	−0.52	−1.47**	−1.53**	−2.65**	
P2×P4	−0.63	−2.18*	−6.13*	−9.20**	0.35	0.28	−0.20	−1.09*	
P2×P5	0.40	0.36	0.09	−1.84	−1.08**	−0.99*	−1.80**	−1.10*	
P2×P6	1.57	2.59*	−1.25	−4.46	0.26	0.71	0.38	−0.29	
P2×P7	3.80**	1.86	−3.96	−2.73	0.08	0.25	0.77	−1.73**	
P2×P8	2.27*	0.89	−2.92	−0.15	0.30	0.42	−0.83	−0.93*	
P3×P4	−1.16	0.12	3.59	0.30	−0.29	−0.87	0.16	−0.23	
P3×P5	1.54	1.32	2.30	2.70	0.90*	1.38**	−0.20	−0.98*	
P3×P6	1.70	−0.11	−4.34	−0.72	−0.03	−0.14	−0.82	−1.80**	
P3×P7	−1.06	−0.18	−2.97	−0.75	0.21	0.65	−0.78	0.53	
P3×P8	1.07	−0.81	6.54**	4.08	0.64	0.41	0.89	1.53**	
P4×P5	0.14	1.42	−4.82*	−2.60	0.61	0.46	0.69	0.64	
P4×P6	−0.36	−2.68*	2.74	2.10	0.78*	0.41	−0.25	0.12	
P4×P7	−3.80**	−4.74**	−0.68	1.89	0.36	0.04	0.15	0.88*	
P4×P8	0.00	2.29*	4.85*	5.50*	0.66	0.39	0.63	1.22**	
P5×P6	0.67	0.52	3.75	5.62*	0.59	1.09*	1.30*	1.53**	
P5×P7	−1.43	−0.54	2.78	5.97*	0.11	0.44	0.72	0.61	
P5×P8	0.70	0.49	−7.56**	−4.82*	0.53	−0.01	0.14	0.14	
P6×P7	−0.26	1.02	7.78**	4.87*	0.58	0.56	0.99	1.35**	
P6×P8	−3.13**	−2.28*	4.98*	1.15	−0.78*	−1.21*	−0.98	−0.18	
P7×P8	−0.23	−0.34	0.83	1.46	−0.01	−0.26	−0.24	−0.42	
LSD5%(sij)	1.80	2.12	4.69	4.65	0.72	0.95	1.08	0.87	
LSD1%(sij)	2.39	2.82	6.23	6.16	0.96	1.25	1.43	1.15	
Cross	Number of spikes/plant	Number of grains per spike	1,000-grain weight (g)	Grain yield per plant (g)	
NOR	WAD	NOR	WAD	NOR	WAD	NOR	WAD	
P1×P2	0.61	2.42*	2.41	2.36	4.53**	4.80*	2.68	6.79**	
P1×P3	−1.44	−0.57	−1.12	1.93	0.12	3.36	−3.96*	2.30	
P1×P4	−1.12	0.42	−1.31	−2.04	−0.33	0.59	−1.78	−1.98	
P1×P5	0.92	1.76	2.97*	−6.24**	2.83	−1.72	9.10**	6.27**	
P1×P6	−2.79*	−1.10	−3.26*	2.06	−6.78**	−5.83**	−5.05**	−5.88**	
P1×P7	−0.86	−3.04**	−8.09**	−8.47**	−2.27	−3.91*	−4.78*	−2.24	
P1×P8	0.94	−0.10	−0.16	1.40	3.60*	−3.97*	−2.35	−6.15**	
P2×P3	−1.56	−1.91	−5.66**	−5.20**	−2.51	2.01	−4.69*	−5.76**	
P2×P4	−0.12	−1.37	1.16	−2.50	1.98	−1.66	6.32**	−1.99	
P2×P5	0.55	−1.66	0.77	−1.37	3.33	−1.87	1.53	−2.04	
P2×P6	−0.26	0.72	−5.46**	0.60	3.93*	0.36	−0.44	1.87	
P2×P7	−1.88	−1.73	−0.29	−6.94**	0.32	−2.09	−0.51	−1.17	
P2×P8	−0.75	−2.46*	−2.02	−1.40	−1.53	−1.58	0.61	−1.14	
P3×P4	0.64	1.62	−1.37	−2.60	2.49	−7.90**	−4.95*	−5.73**	
P3×P5	−0.72	−1.14	−1.76	−2.80	−1.79	−1.24	2.32	4.94*	
P3×P6	1.20	2.40*	−1.99	−5.84**	5.05**	2.85	0.98	0.57	
P3×P7	−0.74	−0.68	−1.82	1.96	−0.01	−1.07	2.47	−2.33	
P3×P8	2.89*	2.28*	9.44**	8.50**	3.63*	7.71**	7.88**	8.79**	
P4×P5	−1.33	0.16	2.84*	5.90**	1.86	4.19*	4.28*	5.50**	
P4×P6	4.96**	0.07	1.16	3.20*	−3.03	2.41	2.31	3.90	
P4×P7	0.03	0.42	−0.34	−1.67	1.61	5.46**	−4.08*	3.27	
P4×P8	−2.45*	0.19	0.26	1.53	0.11	−2.26	1.55	1.81	
P5×P6	−1.84	−2.82*	1.10	−0.67	0.68	1.61	−0.20	−3.30	
P5×P7	1.79	1.10	1.56	2.80	−2.08	−1.91	1.03	2.10	
P5×P8	0.05	−0.38	−2.13	0.33	−4.81**	−1.27	−2.10	3.38	
P6×P7	0.46	1.48	4.05**	3.76**	−0.29	1.25	4.57*	9.73**	
P6×P8	−2.08	−3.03**	1.98	−1.70	3.71*	−5.57**	−2.98	−2.06	
P7×P8	2.52*	1.19	−1.52	−0.57	0.17	4.17*	1.22	−4.02*	
LSD5% (sij)	2.32	2.21	2.49	2.81	3.40	3.86	3.78	3.96	
LSD1% (sij)	3.08	2.93	3.30	3.73	4.50	5.12	5.01	5.25	
Note:

* and ** imply significance at the 0.05 and 0.01, respectively.

Heterosis

The heterosis values for the cross combinations are shown in Table S3. The maximum negative and significant Better-Parent Heterosis (BPH) for days to heading, towards earliness, was recorded by P4×P7 under both conditions. The hybrid P1×P7 exhibited desirable heterotic effects for plant height under normal conditions, while P1×P5 and P1×P8 showed favorable heterotic effects under stress conditions. The strongest positive significant heterotic effects for the spike length were recorded by P4×P8 under normal, and P3×P5 and P5×P6 under drought stressconditions. For number of spikes/plant, the top significant value was recorded by P4×P6 under normal conditions. The hybrid P3×P8 displayed the pronounced significant heterotic effects for number of grains per spike under both treatments. In terms of 1,000 grain weight, P1×P2 under normal and P4×P7 under drought stress conditions recorded the largest significant heterotic. The crosses P2×P5 under normal, P5×P7 and P6×P7 under drought conditions, and P1×P5 and P3×P8 under both conditions showed the greatest BPH heterosis for grain yield.

The relationship among evaluated genotypes and measured traits

The PCA was used to investigate the relationships among the traits studied under drought stress. The first two PCAs explained 62.15% of the variance (40.19% from PCA1 and 21.96% from PCA2), leading to the generation of the PCA biplot (Fig. 6A). PCA1 differentiated the genotypes into positive and negative sides. The evaluated traits were associated with the genotypes on the positive side of PC1, indicating that these genotypes, particularly P3×P8, P5×P8, P7×P8, P7, P5 and P8, demonstrated high agronomic performance. Conversely, genotypes located on the negative side of PCA1, such as P2 and P2×P7, exhibited lower agronomic performance. A strong positive correlation was observed between grain yield and each of plant height, number of grains per spike, number of spikes per plant, and 1,000-grain weight. The heatmap analysis, using a color scale under drought conditions, showed high values of agronomic traits in red and low values in blue (Fig. 6B). Genotypes P3×P8, P5×P8, P7×P8, P7, P5 and P8 were highlighted with superior values for most agronomic traits (red), whereas P2, P2×P4, and P2×P7 exhibited the lowest values (blue) under drought stress conditions.

Figure 6 Principal component biplot (A), and heatmap and hierarchical clustering of 28 evaluated wheat F1 crosses alongside their parental genotypes, based on the evaluated traits under water deficit conditions.

In the heatmap, the values indicating high performance are presented in red colour, while blue colour indicates lower performance. SL: spike length, Spikelets: Number of spikelets, Grains: Number of grain per spike, DTH: days to heading, No. of spikes per plant, and Yield: Grain yield per plant, PH: plant height, and Spikes: TGW: 1,000 grain weight.

Discussion

Breeding wheat cultivars with high yield and drought tolerance is crucial for maintaining productivity amid climate shifts and increasing food demands. This study evaluated the genetic variation of local and imported bread wheat genotypes under normal and drought-stress environments. Stress timing occurred during critical growth stages of heading and grain filling, as these stages are most sensitive to water availability and directly influence yield components. Therefore, the genotypes with enhanced performance likely possess adaptive traits for maintaining under drought stress. The obtained results demonstrated substantial differences within the tested parents, and their crosses under irrigated and drought-stress environments across all traits analyzed. This underscores the presence of adequate diversity that can be utilized for developing drought-resistant genotypes. This aligns with recent findings, such as those by Fatanatvash et al. (2024), Guizani et al. (2024), Sommer et al. (2023), Yang et al. (2023), which demonstrated high genetic variations across multiple agronomic in bread wheat genotypes subjected to drought stress.

Simple sequence repeats (SSRs) markers are valuable for examining genetic differences within bread wheat genetic materials. In this study, number of amplified bands/alleles ranged from 1 to 5, averaging 1.83 alleles per locus. These results are consistent with those of Belete et al. (2021), Guizani et al. (2024), Haque et al. (2021), Türkoğlu et al. (2023), which documented a comparable number of alleles in wheat. The employed markers uncovered high genetic diversity value of 0.53, indicating notable genetic variation among the considered genotypes.

Previously published reports highlighted the utility of SSR markers in evaluating genetic diversity. Galal et al. (2023), Mammadova et al. (2024a), Patel, Patel & Tomar (2024), Su et al. (2023), Verma et al. (2024), and Ezzat et al. (2024) demonstrated the effectiveness of SSR markers in distinguishing genetic diversity in wheat genotypes. The minimal genetic distance was detected between the two imported genotypesLine-123 (P5) and Line-121 (P8), suggested that these genotypes may share common ancestry or similar adaptive responses, which could be leveraged to select core breeding lines for drought tolerance (Ghazy et al., 2024; Kamara et al., 2021). In contrast, the highest genetic distance was observed between the local cultivar Gemmeiza-7 (P7), and the exotic line-125 (P6), suggesting a more pronounced genetic differentiation between these parental lines and, consequently, their offspring. SSR markers effectively categorized the genotypes into two distinct groups, highlighting their utility in differentiating closely related wheat genotypes. By choosing genetically diverse parents from these groups, breeders can develop crosses with better trait combinations and increased heterosis under drought conditions.

Drought stress significantly diminished all measured traits compared to well-irrigated conditions. The decline in plant height is likely due to restricted water uptake, leading to decreased cell growth and division (Ahmad et al., 2018; Mansour et al., 2023; Mousa et al., 2024). The notable decrease in yield traits under drought conditions may be attributed to factors such as pollen sterility, decreased grain set and development, and limited sink capacity, which ultimately led to a lower grain yield (Farooq et al., 2009; Li et al., 2024b; Zhao et al., 2020). The parental genotypes and their hybrid combinations were classified into five categories (A–E) according to their tolerance to drought stress, ranging from highly tolerant to highly sensitive. Crosses P1×P5, P5×P6, P6×P8, and P7×P8 were noted as drought-resistant, demonstrating better agronomic performance than sensitive ones. These resilient crosses offer promising opportunities for incorporation into wheat improvement programs focused on enhancing grain yield in drought-prone environments. Similarly, numerous studies have used cluster analysis and stress susceptibility indices for classifying wheat genotypes subjected to drought stress (Kamara et al., 2024; Mansour et al., 2021; Moustafa et al., 2021; Poggi et al., 2024).

The molecular variation revealed through SSR markers provided considerable genetic diversity among wheat genotypes, which is a critical foundation for developing adapted wheat cultivars (Tyagi et al., 2021). Furthermore, the distinct clustering of genotypes based on drought tolerance reflected the underlying genetic factors that influence agronomic performance under water-deficit conditions. Specifically, genotypes with higher molecular diversity often harbor alleles associated with adaptive traits. These genetic factors directly translate into improved agronomic traits, such as grain yield stability, reduced plant height, and greater 1,000-grain weight under drought stress (Gupta et al., 2020). Future research could be strengthened by performing genome-wide association studies (GWAS) or identifying QTLs related to the key agronomic traits under drought stress. This would provide a more direct understanding of the genetic basis for drought tolerance, bridging the gap between molecular markers and phenotypic outcomes.

Parents with high GCA effectively transmit alleles to their offspring, which is essential for trait enhancement (Sakran et al., 2022; Salem et al., 2020). The current study recognized the local parental cultivar Sids-12 (P2) as an excellent combiner for reducing plant height and achieving earlier maturity under both irrigation treatments, indicating its potential for developing dwarf and early maturing genotypes. The traits of early heading and shorter plant height might function as adaptive mechanisms to avoid terminal drought stress (Galal et al., 2023). Additionally, exotic parental genotypes Line-117 (P3), Line-144 (P4) and Line-123 (P5), exhibited high GCA for grain yield, and some of its attributes under stress conditions. These genetic lines show potential for enhancing bread wheat yields in drought conditions. They could be used to introduce beneficial alleles into progeny and could effectively combine with other genotypes to produce high-performing offspring. These observations align with findings stated by Kamara et al. (2021), Semahegn et al. (2021), and Shamuyarira et al. (2023).

Crosses with notable SCA are excellent candidates used for identifying novel segregates (Kamara et al., 2021). The crosses; P1×P5, P3×P8, P4×P5, and P6×P7 stand out as distinct combiners for boosting grain yield and its associated traits. These crosses, resulting from crosses between strong and weak general combiners, could increase heterosis and yield high-performing genotypes in drought conditions. This could attributed to one parent offering significant additive effects while the other contributes epistatic influences (Ashraf et al., 2015; Mwadzingeni, Shimelis & Tsilo, 2018). Notably, the crosses P1×P5, and P3×P8 showed strong positive SCA and hybrid vigor advantages for grain yield, and related attributes, suggesting that these combinations might yield promising segregants with improved traits in both optimal and drought conditions. Additive and non-additive genetic effects played a crucial role in the inheritance of the traits under investigation, as evidenced by the pronounced GCA and SCA effects. The ratio of GCA to SCA was consistently below one for all traits, revealing the strong contribution of non-additive gene effects in controlling their inheritance. This coincides with the results of Farshadfar, Rafiee & Hasheminasab (2013), Mwadzingeni, Shimelis & Tsilo (2018). However, it contrasts with previous studies of Kamara et al. (2022), and El-Maghraby et al. (2005), which emphasized the predominance of additive genetic effects in the inheritance of grain yield under drought conditions.

SCA values, which represent non-additive genetic effects, showed a strong association with hybrid performance for key agronomic traits under drought-stress conditions. Crosses with high SCA values, such as P1×P5, P3×P8, P4×P5, and P6×P7, consistently exhibited superior grain yield and related traits. This indicates that the interaction between the parental genotypes contributed significantly to hybrid vigor, particularly for traits influenced by dominance and epistatic effects (El Hanafi et al., 2022). A notable association was observed between SCA estimates and grain yield, suggesting that non-additive gene action plays a critical role in determining hybrid performance under water-limited conditions. Crosses involving genetically diverse parents, as indicated by their SSR-based genetic profiles, exhibited higher SCA values, which likely contributed to the enhanced drought tolerance and overall performance of their progeny (Al-Ashkar et al., 2020).

Non-additive gene action was found to predominantly influence grain yield and related traits, particularly under drought-stress conditions. This aligns with findings by Semahegn et al. (2021), Kamara et al. (2021), Mia et al. (2017) who demonstrated that traits such as grain yield and contributing traits are significantly affected by dominance and epistatic effects in water-limited environments. Similarly, Gaballah et al. (2022), El-Mowafi et al. (2021), and Ahmad & Gupta (2024) highlighted that crosses with high SCA values exhibited superior yield performance under stress conditions, underscoring the importance of exploiting non-additive genetic effects in breeding programs.

Conclusions

This study provides valuable insights into the genetic diversity and combining ability of wheat genotypes under drought stress, highlighting the significant potential of specific genotypes and crosses as valuable resources for breeding programs to enhance wheat resilience to climate change. SSR markers effectively exhibited genetic diversity among wheat genotypes to identify promising parental lines for drought tolerance. The local cultivar Sids-12 (P2) demonstrated a strong combining ability for producing shorter and earlier-maturing genotypes, making it a valuable resource in wheat breeding programs. Similarly, the exotic parental genotypes Line-117 (P3), Line-144 (P4), and Line-123 (P5) were identified as exceptional combiners for enhancing grain yield and related agronomic traits. Among the crosses, P1×P5, P4×P5, P3×P8, and P6×P7 stood out as top-performing specific combinations, exhibiting superior grain yield and associated traits under both optimal and drought-stressed (water-restricted) conditions. The identified drought-tolerant genotypes could be employed as foundational material for developing high-yielding, drought-tolerant cultivars, addressing the challenges posed by water-deficit conditions in arid and semi-arid regions. Positive correlation was detected between grain yield and traits such as number of spikes per plant, plant height, 1,000-grain weight, and the number of grains per spike. These traits are particularly valuable for early indirect selection in drought-prone environments due to their ease of measurement, offering practical markers for identifying drought-resilient genotypes. Future research could build on these findings by investigating the molecular mechanisms underlying drought tolerance. Additionally, integrating genomic selection and marker-assisted breeding approaches could accelerate the development of drought-tolerant wheat varieties.

Supplemental Information

Supplemental Information 1 Supplementary materials.

Supplemental Information 2 Raw data of phenotyping field trials.

Supplemental Information 3 Uncropped gel photos.

Supplemental Information 4 Length uncropped gels.

Additional Information and Declarations

Competing Interests

The authors declare that they have no competing interests.

Author Contributions

Mohamed I. Motawei conceived and designed the experiments, performed the experiments, analyzed the data, authored or reviewed drafts of the article, and approved the final draft.

Mohamed M. Kamara conceived and designed the experiments, performed the experiments, analyzed the data, prepared figures and/or tables, authored or reviewed drafts of the article, and approved the final draft.

Medhat Rehan conceived and designed the experiments, performed the experiments, prepared figures and/or tables, and approved the final draft.

Data Availability

The following information was supplied regarding data availability:

The raw measurements are available in the Supplemental Files.

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
