# Peer review of "Exploring molecular variation and combining ability of local and exotic bread wheat genotypes under well-watered and drought conditions"

_PeerJ, doi:10.7717/peerj.18994_

## Round 0.1 · original submission · Major Revisions

You need to strengthen the discussion. Re-interpret your result in relation to the genetic analysis of the diallel analyses and SSR.

Reviewer 1 ·

Basic reporting

Thank you for considering me to review the manuscript "Exploring Molecular Variation and Combining Ability of Local and Exotic Bread Wheat Genotypes Under Normal and Water-Limited Conditions". The manuscript analyzed the genetic diversity, combining ability, and performance of local and exotic bread wheat genotypes under normal and water-limited conditions. The research integrated molecular markers (SSR), diallel mating design, and field evaluations to investigate drought tolerance and yield-related traits, making it valuable to wheat breeding for drought resilience. The study is well-designed and addresses an important topic in wheat breeding, especially drought tolerance. However, some areas of the manuscript require clarification and further explanation to enhance readability and improve scientific rigor. Addressing these points will strengthen the manuscript readability and ensure that the findings are presented robustly and scientifically.

Suggestions:
Title: The title is appropriate and presents the main aim of the study.
Abstract:
The abstract summarizes the main objectives, methodology, and findings but could be more concise. Key results, such as the significant findings on genetic diversity, combining ability, and performance differences between genotypes, should be clearly stated in a more compact form. The abstract could be reduced to approximately 250 words, focusing on the most important findings, particularly the significance of the drought tolerance index and the use of SSR markers. The term "grouping into clusters" could be clarified to emphasize that it refers to drought tolerance.

Introduction:
The introduction provides a background and justification for the study. However, some sections can be more concise, and references to specific research could be better integrated to improve the flow. The introduction provides sufficient context, but some references to recent literature, such as those from Fatanatvash et al. (2024), Guizani et al. (2024), Sommer et al. (2023), and Yang et al. (2023), are mentioned in the discussion but not in the introduction. Consider adding a sentence or two on how recent studies have utilized molecular markers like SSR and how these contribute to understanding drought tolerance in wheat. The long sentences describing the challenges of drought and climate change can be broken down for better readability.

Materials and Methods:
The methodology is well-detailed but could benefit from additional clarity in certain sections.
The section on SSR marker analysis could benefit from information on the protocol used for DNA extraction, amplification conditions, and the type of gel electrophoresis or other methods employed for detecting polymorphisms.
Expand on the environmental conditions, such as temperature and humidity, to give more background to the field trials.
The statistical methods used are appropriate for the study, but providing more detail on the software used. Please provide additional information about how the statistical tests were performed in the software, especially for cluster, principal component, and heatmap analyses. A brief explanation of how GCA and SCA were calculated using Griffing’s Method 2 could be helpful for readers unfamiliar with the procedure.

Results:
The results are well-organized and informative but some areas need further clarification. For example, categorizing genotypes into clusters based on drought tolerance is a significant result. More information on how these clusters were defined and the statistical methods used to determine the grouping would be beneficial.
The figures are essential for visualizing data but should be more precise and detailed. Ensure that figure legends clearly describe what is being presented and what each cluster or category represents. Ensure all graphs are labeled with units of measurement (e.g., days, cm, kg, etc.). Figure 6 (PCA Heatmap): The heatmap is helpful, but a more precise explanation of the color scale and the relationship between the values and the genotypes would help readers. The dendrogram and heatmap could benefit from clearer labeling.
The tables are well-organized but could use some clarification. Table 4 (Analysis of Variance) presents a lot of information. Adding bolded or italicized values to highlight significant differences may be helpful.

Discussion:
Compare your findings thoroughly with the literature, providing greater insight into the molecular and agronomic results.
The linkage between molecular variation and agronomic traits could be discussed more thoroughly. How does the observed molecular variation directly contribute to the drought-tolerant traits identified?
The relationship between SCA results and hybrid performance could be explored in more detail. Are there any significant correlations or interactions between these factors?
While the manuscript mentions the role of non-additive gene action in yield-related traits, a deeper discussion on how these findings compare with previous studies would strengthen the argument.

Conclusion:
The conclusion summarizes the study main findings well but could be more forward-looking in terms of future research. Emphasize potential applications of the study in wheat breeding programs. Suggest future research directions to address any limitations or to build on the findings, such as exploring the molecular mechanisms behind drought tolerance. FOR EXAMPLE, This study provides valuable insights into the genetic diversity and combining ability of wheat genotypes under drought stress. The identified drought-resistant genotypes and hybrids can serve as useful resources for breeding programs to improve wheat resilience to climate change. Future research could explore the underlying molecular mechanisms of drought tolerance and the integration of genomic selection in wheat breeding.

References:
Ensure all cited studies (such as those by Fatanatvash et al., Guizani et al., and Sommer et al.) are correctly listed in the references section. Double-check all citations for accuracy.

Language:
The manuscript is written in clear scientific English, but there are a few areas where readability can be improved. Break down complex sentences for easier reading and comprehension. Ensure consistent use of terms (e.g., "drought stress" vs. "water-limited conditions").

Experimental design

NA

Validity of the findings

NA

Additional comments

NA

Reviewer 2 ·

Basic reporting

The paper entitled "Exploring molecular variation and combining ability of local and exotic bread wheat genotypes under normal and water-limited conditions" deals with a pre-breeding issue which aims to identify the most approriate parents for cross to generate new varaiability and superior genotypes.

In general the introduction sets the research in its context. However some sentences should be revised like that in lines 55-56 "In Egypt..." and replaced by rainfall data related to crops and not to human use.

I suggest to replace some terms like "alley" and "seepage" by others more commonly used.

In general the language needs to be improved and more adequate scientififc terms should used. For instance, In line 128, "panicle" should be replaced by "spike".

Experimental design

The Material and Methods section should be improved. It is better to subdivise "Molecular Profiling of Parental Genotypes" into plant material subsection and Molecular profiling. More details are required for the genotypes used in the study, regarding their history in Egypt, tolerance to drought stress and their overall agronomic performance under Egyptian conditions. In addition, the number of genotypes should be mentined for parents and offsprings.

The Twelve SSR markres were selected according to their relation to drought tolerance in wheat. The traits with which the markers were correlated should be mentioned for exemple grain yield, RWC... Moreover, the original references (marker-trait Relationships) should be cited.

In 2022-2023 season, the F1 offspring are in segregation (F2), where each signle seed could give different plant. How did you managed to have three replication?

Information about the soil humidity should be provided, and the type of stress, timing should be discussed based on the results.

Validity of the findings

'no comment'

·

Basic reporting

The manuscript entitled “Exploring molecular variation and combining ability of local and exotic bread wheat genotypes under normal and water-limited conditions" presents a well-structured and comprehensive study addressing a critical agricultural challenge by developing drought-tolerant wheat genotypes. Integrating molecular marker analysis and field evaluation under different irrigation conditions provides robust insights into the genetic diversity of wheat genotypes. The research is timely and highly relevant, considering the increasing severity of climate change and its impact on global food security. The authors have successfully demonstrated the utility of SSR markers in evaluating genetic diversity and highlighted the significance of non-additive gene actions in improving grain yield under drought stress. Furthermore, the study provides practical breeding implications by identifying specific parental lines and hybrids with superior performance, which could significantly contribute to sustainable wheat production in water-limited environments. Overall, this manuscript makes a valuable contribution to wheat breeding and drought tolerance research and is suitable for publication.

Experimental design

The materials and methods section is well-detailed, offering clear and replicable descriptions of the experimental design, SSR marker selection, and statistical analyses.

Validity of the findings

The manuscript can benefit from minor improvements to enhance clarity and reader engagement.
Minor revisions
Line 23: Use consistent terms for F1 hybrids or "F1 crosses"
Line 37: "...due to their ease of measurement"—remove the comma before "measurement."
The introduction provides a strong background, but consider removing redundancy regarding drought effects on wheat production, as this is extensively covered.
The materials and methods section is well-detailed, offering clear and replicable descriptions of the experimental design, SSR marker selection, and statistical analyses.
The results are well-written, presenting comprehensive findings supported by appropriate statistical analyses and graphical illustrations.
Ensure all figures and tables are referenced sequentially in the text.
Add clearer legends to Figures 5 and 6, explaining abbreviations and clusters for accessibility to broader audiences.
The discussion integrates the results with relevant literature, providing meaningful insights into the genetic mechanisms underlying drought tolerance and the practical implications for wheat breeding. Identifying specific high-performing parental genotypes and hybrids under water-limited conditions is particularly noteworthy. Highlight practical implications of identifying drought-tolerant hybrids like P1×P5 and P3×P8 for breeders. Discuss potential applications of the identified genotypes beyond drought-stress environments, if applicable.
Include a brief comparison of findings with prior studies to better situate the study within the existing literature.
Ensure that all citations in the text are correctly formatted to match the reference list.

---

## Round 0.2 · accepted · Accept

Your corrections is adequate for acceptance.

Reviewer 1 ·

Basic reporting

Authors have revised the manuscript accordingly and it is well received.

Experimental design

No further comments.

Validity of the findings

No further comments

Additional comments

No further comments

Reviewer 2 ·

Basic reporting

The authors have successfully revised the manuscript according to the suggestions.

Experimental design

'no comment'

Validity of the findings

'no comment'

Additional comments

'no comment'

·

Basic reporting

.

Experimental design

.

Validity of the findings

.

Additional comments

The authors have addressed my prior concerns.